# Modesty and Security: Attributes Associated with Comfort and Willingness to Engage in Telelactation

**DOI:** 10.3390/children8040271

**Published:** 2021-04-01

**Authors:** Adetola F. Louis-Jacques, Ellen J. Schafer, Taylor A. Livingston, Rachel G. Logan, Stephanie L. Marhefka

**Affiliations:** 1Department of Obstetrics and Gynecology, Morsani College of Medicine, University of South Florida, Tampa, FL 33606, USA; alouisjacques@usf.edu; 2College of Nursing, University of South Florida, Tampa, FL 33612, USA; 3Department of Community and Environmental Health, College of Health Sciences, Boise State University, Boise, ID 83725, USA; ellenschafer@boisestate.edu; 4Department of Anthropology, University of Nebraska—Lincoln, Lincoln, NE 68588, USA; tlivingston3@unl.edu; 5College of Public Health, University of South Florida, Tampa, FL 33612, USA; logan23@usf.edu; 6The Chiles Center, University of South Florida, Tampa, FL 33613, USA

**Keywords:** breastfeeding, telemedicine, lactation support, telelactation

## Abstract

The objectives were to identify conditions under which mothers may be willing to use telelactation and explore associations between participant characteristics, willingness, and beliefs regarding telelactation use. Mothers 2–8 weeks postpartum were recruited from two Florida maternal care sites and surveyed to assess demographics, breastfeeding initiation, and potential telelactation use. Analyses included descriptive statistics and logistic regression models. Of the 88 participants, most were white, married, earned less than USD 50,000 per year, had access to technology, and were willing to use telelactation if it was free (80.7%) or over a secure server (63.6%). Fifty-six percent were willing to use telelactation if it involved feeding the baby without a cover, but only 45.5% were willing if their nipples may be seen. Those with higher odds of willingness to use telelactation under these modesty conditions were experienced using videochat, white, married, and of higher income. Mothers with security concerns had six times the odds of being uncomfortable with telelactation compared to mothers without concerns. While telelactation can improve access to critical services, willingness to use telelactation may depend on conditions of use and sociodemographics. During the COVID-19 pandemic and beyond, these findings offer important insights for lactation professionals implementing virtual consultations.

## 1. Introduction

Breastfeeding is beneficial for infants and mothers [1,2,3]. As such, the American Academy of Pediatrics and the American College of Obstetricians and Gynecologists recommend exclusive breastfeeding of infants for six months and supplementation with complementary foods for at least one year [2,4]. Many women are unable to meet these recommendations and their breastfeeding goals due to challenges [5,6] including lack of social support, perceived insufficient milk supply, and latch difficulty [7,8]. Breastfeeding support services provided by a skilled lactation provider can address these challenges [9]. However, access to skilled lactation services is not geographically equitable in the United States [10]. For example, Florida, the third most populous state in the nation and the setting of this study, falls well below the recommended and national average of International Board Certified Lactation Consultants (IBCLC). While the recommended number of IBCLCs is 8.6 per 1000 births, Florida has only 2.55, compared to the national average of 3.86 per 1000 births [11].

Research has shown that telelactation, or the use of face-to-face videoconferencing software for lactation assistance via smartphone, tablet, or computer, is a promising option to provide professional lactation support [12,13,14,15,16,17,18,19,20], particularly for women in rural areas [15,19,20]. However, there are few studies on the acceptability of telelactation support [13,14,17], and no studies on the acceptability of this technology among urban women from diverse demographic backgrounds. The primary objective of this study was to characterize technology use among urban mothers during the postpartum period and determine conditions under which women may be willing to adopt telelactation services and support. The secondary objective was to explore how participant characteristics impact willingness and beliefs regarding use of telelactation services. Data on conditions of use and sociodemographic differences in acceptability of telelactation services are particularly salient given the current SARS-CoV-2 pandemic as many women may be unwilling or unable to receive in-person lactation services.

## 2. Materials and Methods

### 2.1. Procedures

English- or Spanish-speaking women who were at least 18 years old and presenting for postpartum follow-up (between 2 and 8 weeks postpartum) were recruited to participate in the study at two maternal care sites in Tampa, FL. One recruitment site served mostly Medicaid patients while the other typically served those with private insurance, allowing a range of socioeconomic backgrounds. Women who did not have a live birth, whose children were admitted into the neonatal intensive care unit, or who were separated from their infants (e.g., foster care, adoption) were excluded from the study.

Clinic staff distributed survey packets to eligible participants. All contents of the packet were at a fifth-grade reading level and were available in English and in Spanish. Patients were instructed to read the contents of the survey packet, which included an introductory letter, informed consent, survey, future contact form, and gift card information. They were advised to contact research staff regarding any questions. Participants had the option to complete the survey via paper or online during the clinic visit or at a later time. The clinic staff member collected survey packets in sealed envelopes and placed the envelopes in a secure location. Participants who completed the survey via paper at a later time mailed the paper survey to the study office. Participants who chose to complete the survey online entered a link or scanned a QR code provided in the survey packet and completed all forms online.

The front desk staff distributed the survey packets to eligible participants (see File S1: “What Do Moms Need” Study Protocol). A total of 114 mothers provided consent and participated in a self-administered survey, receiving a USD 10 gift card for their time. Data were collected from November 2015 to June 2016. For the purpose of this study, only those who initiated breastfeeding and completed all survey questions (N = 88) were included in the analysis. All subjects gave their informed consent for inclusion before answering survey questions. The study was conducted in accordance with the Declaration of Helsinki, and the protocol was deemed exempt by the University of South Florida’s Institutional Review Board (Pro000023145). 

### 2.2. Measures

#### 2.2.1. Breastfeeding Initiation

Mothers were defined as having initiated breastfeeding, and thus included in these analyses, if they responded “yes” to any combination of the following prompts: (1) In the hospital, I tried to feed my baby from the breast, (2) In the hospital, I tried to pump breast milk, or (3) In the hospital, my baby received some breast milk from me. 

#### 2.2.2. Willingness to Use Telelactation Services

Willingness to use telelactation was assessed by having mothers read a brief synopsis of a theoretical breastfeeding support application that would connect mothers to lactation professionals (Figure 1: You and Technology) and then asked mothers to rate their willingness to use their mobile phones for this application through several Likert scale questions (1 = definitely not to 5 = definitely yes). Questions were developed with the goal of providing feedback for the purposes of developing a telelactation app. Questions were developed by the study team and reviewed for clarity and content by lactation professionals and obstetricians. The four questions used in this analysis were: “Would you be willing to use this professional breastfeeding support program if…” (1) while breastfeeding the professional may see your nipples via videochat? (2) it involved the professional seeing you feeding your baby without a cover? (3) you knew it was over a secure server? (4) the whole program was free (Table 1)? These four items were dichotomized for analysis (probably yes or definitely yes vs. any other response).

#### 2.2.3. Beliefs Regarding Use of Telelactation Services

Mothers were also asked about their beliefs regarding the telelactation application on their mobile devices with the following prompt: “The next set of questions ask about your feelings about using your mobile phone for professional breastfeeding support, just like you read about earlier. Please respond to each item below.” Each of the following items were presented, and participants were asked to pick an option from a Likert-type scale (1 = strongly disagree to 5 = strongly agree): (1) Videochat is good for some things, but not for getting help with breastfeeding. (2) If you use video-based breastfeeding support over a computer or Internet you can never be sure who will have access to the videochat session. (3) Videochat can provide access to breastfeeding support wherever women need it. (4) Women will not develop a personal relationship with a breastfeeding support professional if they only interact via videochat. (5) I do not feel comfortable with accessing video-based breastfeeding support online. These items were adapted from the Technology Readiness Index [21] and the Technology Readiness Index 2.0 [22] and were dichotomized for analysis (1 = agree or strongly agree, 0 = any other response). Other questions were presented in the survey but were not the focus of this analysis. 

### 2.3. Participant Characteristics

We examined characteristics of the participants, including access to technology (computer, tablet, mobile phone with text messaging, and smart mobile phone); items were derived from the PEW Internet and American Life Project [23]. Past experience using videochat, age, marital status (married vs. any other response), employment status (working full- or part-time or self-employed vs. any other response), income (at least USD 50,000 per year vs. any other response), race (Black, white), and ethnicity (Hispanic vs. any other response) were assessed using investigator developed questions. 

## 3. Analyses

Participant characteristics were explored using descriptive statistics. Separate bivariate logistic regression models were used to assess associations between participant characteristics and each of the nine outcome variables related to willingness to use telelactation and beliefs regarding telelactation services. To further explore mothers’ comfort with accessing telelactation, the factors significantly associated in bivariate analysis with mothers agreeing to the statement “I do not feel comfortable with accessing video-based breastfeeding support online” were analyzed with multivariable logistic regression. A final model was derived by backward elimination. Lastly, we conducted a post hoc descriptive analysis to describe mothers who were simply not willing to use telelactation under any of the four conditions we asked about (i.e., if it’s free, if nipples show, if nursing without a cover, and if over a secure server). 

## 4. Results

Eighty-eight women who initiated breastfeeding and completed relevant items on the survey were included in the analyses. Most mothers were married, not currently working or self-employed, earned less than USD 50,000 per year, and identified as white. On average, participants were 29.3 years old (±5.05) with ages ranging from 18–44 years. A majority of participants had access to each type of technology (computer, tablet, text messaging, smart phone); 94.3% indicated access to a smart phone and 53.4% indicated they had previously used the Internet for videochat. Participant characteristics are in Table 2.

### 4.1. Willingness and Beliefs Regarding Telelactation Use

As shown in Table 2, most participants were willing to use telelactation if the whole program was free (80.7%), if they knew it was over a secure server (63.6%), and if it involved feeding the baby without a cover (55.7%). However, only 45.5% were willing to use telelactation if the professional may see their nipples via videochat. Despite 64.8% of participants agreeing with the belief that videochat can provide access to breastfeeding support wherever women need it, 48.9% of participants agreed with the belief that if you use video-based breastfeeding support via a computer or internet, you can never be sure who will have access. Upon exploring conditions under which women would use telelactation if the whole program was free (Figure 2), we found that 15 women would not use it under any conditions (see note for Figure 2).

### 4.2. Participant Characteristics and Willingness and Beliefs Variables

We explored the willingness to use telelactation, beliefs about telelactation, and each demographic and technology access/use factor in bivariate analyses (see Table 3). Those with higher odds of being willing to use telelactation if the professional may see their nipples or if it meant feeding without a cover tended to be married, in the higher income category, white, and had experience using videochat. Those who indicated Black as their race had lower odds of willingness to use telelactation if it involved breastfeeding without a cover. Those with higher odds of being willing to use telelactation if they knew it was over a secure server tended to be married, in the higher income category, white, and had experience using videochat. Those who were married had greater odds, compared to those not married, to be willing to use telelactation if the program was free. Participants who were older, married, in the higher income category, and white had higher odds of agreeing with the belief that videochat can provide access to breastfeeding support wherever women need. White mothers were less likely to agree with the belief that “women will not develop a personal relationship with the breastfeeding support professional” if receiving breastfeeding support remotely.

### 4.3. Participant Characteristics and Discomfort

To better define the 26.1% of participants who agreed they did not feel comfortable accessing video-based breastfeeding support online, we explored variables related to demographics, access to technology, and the other willingness and belief variables as predictors of not feeling comfortable with telelactation. The full model, in Table 4, includes those items that were associated with discomfort of using telelactation in bivariate analysis (willingness to use telelactation if: the professional may see your nipples, may see you feeding your baby without a cover, agreeing if you use video-based breastfeeding support you can never be sure who will have access, and agreeing videochat can provide access to breastfeeding support wherever women need it). There were no significant associations between discomfort and all demographic factors, all access to technology items, willingness to use telelactation if you knew it was over a secure server, willingness to use telelactation if the program was free, and agreeing that women will not develop a personal relationship with a breastfeeding support professional. The final model, in Table 4, shows mothers who agreed with concerns regarding who may have access to their telelactation session have more than six times the odds (OR = 6.32, 95% CI: 1.82–21.98) of not being comfortable with telelactation compared to mothers who did not agree. Alternately, those who were willing to use telelactation if the breastfeeding professional might see their nipples (OR = 0.20, 95% CI: 0.05–0.89) and agreed that videochat can provide access to breastfeeding support wherever women need it, had lower odds (OR = 0.29, 95% CI: 0.08–0.99) of not feeling comfortable with accessing telelactation support.

### 4.4. Post Hoc Analysis

Fifteen mothers (17%) were not willing to use telelactation under any condition (Figure 2). Most of these 15 mothers were not working (80.0%), had an income lower than USD 50,000 per year (80.0%), had never used videochat before (60.0%), and owned a smart phone (86.7%). The mothers unwilling to use telelactation were also unwilling to use (or had not previously used) lactation support: in a clinic (60%) or via a doula/birth attendant (60%); but most indicated they had used or were willing to use a support group (53%), or a website (73%) or text messaging (73%) for baby-related information.

## 5. Discussion

In this study, we found that most mothers who initiated breastfeeding expressed willingness to use telelactation if it was free (81%). Yet, prior to encountering the “free” question (which came last on the questionnaire), when they were asked about other conditions of telelactation use (if the professional could see the woman’s nipples; if they would need to breastfeed over video without a cover; and if over a secure server), mothers’ willingness was lower. Those more likely to be willing to use telelactation were married, in a higher income bracket, white, and had experience using videochat platforms, suggesting lower-income, non-white participants may have lower uptake of telelactation programs. These results differ from the only other quantitative study of the acceptability of telelactation, which found no significant differences in willingness to use the technology based on sociodemographic factors [14]. Our results suggest willingness to engage in telelactation may be contingent on a multitude of factors.

Other studies indicate that telelactation is a viable alternative to in-person lactation support for some populations [13,14,15,17,20]; however, our findings suggest it may not be acceptable to all. The complexity of acceptability of telelactation is supported by other findings from this study. Approximately 26% of women reported discomfort with the prospect of using telelactation. Greater discomfort was significantly associated with greater concerns about: (1) who may have access to the virtual session, (2) nipple exposure, and (3) disagreeing that telelactation can offer mothers access to support whenever they need it.

In contrast to our findings for willingness, we found that sociodemographic characteristics did not relate significantly to discomfort with telelactation use. It is possible that discomfort around telelactation might transcend sociodemographic groups, but willingness to adopt telelactation varies across sociodemographic groups. Black, lower income, and unmarried women have disparate breastfeeding outcomes and encounter many barriers and challenges in achieving their personal breastfeeding goals [24,25,26,27,28]. Our findings suggest that mothers who are unwilling to use telelactation might overlap with groups at risk for poor breastfeeding outcomes. Similarly, in a study of mothers within Mississippi’s Women, Infant Children (WIC) program, only 7 percent downloaded a free telelactation app [29]. These findings suggest that as lactation services become more available on virtual platforms, it is critical that we find ways to support all breastfeeding mothers so that we do not further widen breastfeeding inequities. Breastfeeding inequities are a public health priority because they may contribute to other inequities in maternal and child health outcomes [28,30]. 

Fifteen mothers were unwilling to use telelactation under any condition. Most of these mothers were unemployed and of lower income, increasing their risk for poor breastfeeding outcomes [26]. Most mothers in this category had a smart phone but had never used videochat. This finding is consistent with another study showing people who had not used certain technologies were less likely to be willing to use those technologies to receive health-related information, even if the technology/service was free and accessible [31].

The mothers unwilling to use telelactation under the conditions investigated in this study were also unwilling to receive lactation support in person; but most were willing to use a website or text-messaging for baby-related information. It is important to understand how to best support this group of mothers who could be at high risk for early breastfeeding cessation [26]. Providing access to text-messaging or online patient-friendly breastfeeding resources such as videos on latch and breastfeeding positions or materials on breastfeeding challenges and management may be helpful to these mothers [32,33,34]. Further research exploring concerns and barriers regarding in person or virtual lactation support could help in development of resources.

The SARS CoV-2 (COVID-19 disease) pandemic and public health guidance on mitigation efforts have created new breastfeeding challenges. For example, the Centers for Disease Control and Prevention (CDC) recommends [35] determining on a “case-by-case basis” if a mother with suspected or confirmed COVID-19 should be separated from her newborn. In addition to the potentially unnecessary [36] separation of the breastfeeding dyad, other stressors may increase the need for lactation support. These stressors include COVID-19 anxiety [37], potential lack of support persons (including doulas) present during and after labor [38,39], possible redirected resources [40] and expedited discharge from the hospital [41] leading to missed breastfeeding education opportunities, as well as social isolation caused by physical distancing [35]. As breastfeeding is an important aspect of public health [42] with lifelong benefits for mother and child [43], it is critical that mothers receive the support they need in this unprecedented time. While many skilled lactation specialists have moved to the virtual realm to provide these necessary support services, our results indicate telelactation is not a “one size fits all” solution.

Indeed, findings suggest that lactation professionals may wish to adjust their practice to accommodate the comfort of each mother. When scheduling a telelactation visit, lactation providers may wish to provide anticipatory guidance—over the telephone, on their website or social media, or through information sent via text or email. This guidance should detail the role of the telelactation professional, as well as the potential activities involved in telelactation. Using the principles of shared decision-making [44], anticipatory guidance should be reviewed with the mother before the session, with plans regarding activities that will be conducted during the session. Ideally, providers will inform mothers that they will determine what happens in the telelactation session, based on their level of comfort and willingness to engage in certain activities (specifically showing their nipples, which is not always required in lactation visits). Additionally, providers should take care to: assure mothers that telelactation sessions will not be recorded or saved on any server, conduct sessions in private rooms (i.e., not in open areas of their home or office where other people may see the telelactation session [45]), and use HIPAA-compliant platforms when feasible [46] (see Table 5 for all recommendations).

Understanding and expanding the use of technology for breastfeeding support even after the COVID-19 pandemic may be important in helping mothers sustain breastfeeding, as research has shown women who used telelactation had increased exclusive breastfeeding and longer durations of breastfeeding than women who only had access to in-person lactation services [20,29]. Women in our study who perceived telelactation as offering support to women “wherever they needed it” were less likely to feel uncomfortable using telelactation, suggesting that marketing this attribute may be effective in increasing uptake. Lactation professionals have a positive effect on breastfeeding exclusivity and duration [47,48], and telelactation, which allows for live counseling to mothers outside of normal clinical hours, offers a less time consuming and more accessible [13] way for women to attain breastfeeding support than in person consultations.

### Strengths and Limitations

This study provides relevant, timely information on how to support breastfeeding mothers using telelactation and points to considerations that may impact initial uptake or sustained engagement with telelactation support. We sampled a diverse group of women receiving care in urban maternal care sites. However, our study has limitations, including a limited sample size and the consequential limited power to detect statistically significant differences. Indeed, the initial goal of the study was to conduct formative research to inform the development of a mobile telelactation app, though we still found significant associations. As such, while a larger sample would allow for greater exploration of willingness to use and comfort with telelactation, the results of this study are still relevant for those who are interested in working with women via telelactation. Additionally, measurement of the construct “willingness” may not fully encompass the barriers, including structural barriers, to women accessing and feeling comfortable using telelactation. Future research should qualitatively explore women’s willingness to use telelactation services in order to ascertain barriers and ways to assuage discomfort with its usage. Further, additional studies are needed to investigate the willingness to use and comfort with telelactation services for women not included in our sample, including those whose infants spend time in the neonatal intensive care unit, separated from the infant, and/or were too ill or unable to initiate breastfeeding. Finally, the data were collected in 2015 and 2016, prior to the COVID-19 pandemic. Mothers may be more willing to use lactation support during the pandemic given transmission concerns regarding non-physically distanced care. However, our results provide relevant, timely information on how to support breastfeeding mothers using telelactation and points to sustained concerns (pandemic or not) that may impact initial uptake or engagement with telelactation support.

## 6. Conclusions

Telelactation is perceived by some as a viable alternative to in-person lactation support; however, it may not be acceptable to all. Future research should identify ways to engage mothers who have concerns with telelactation use and continue efforts to increase access to and comfort with breastfeeding support through technology. Such engagements should particularly focus on innovative ways to support mothers who are uncomfortable using telelactation.

## Figures and Tables

**Figure 1 children-08-00271-f001:**
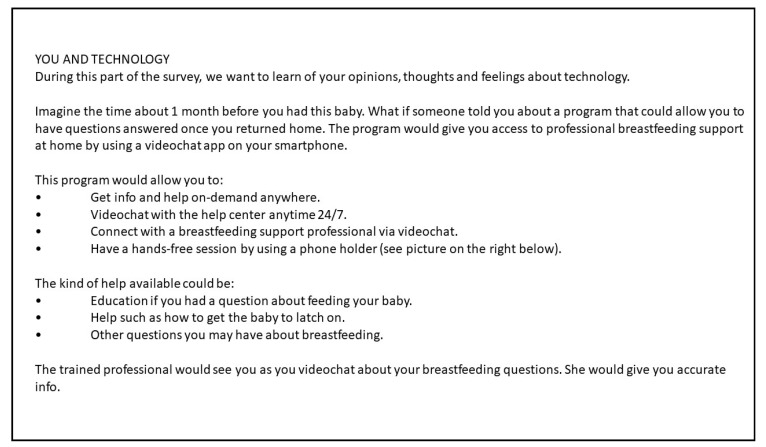
Brief synopsis of breastfeeding support application provided to participants.

**Figure 2 children-08-00271-f002:**
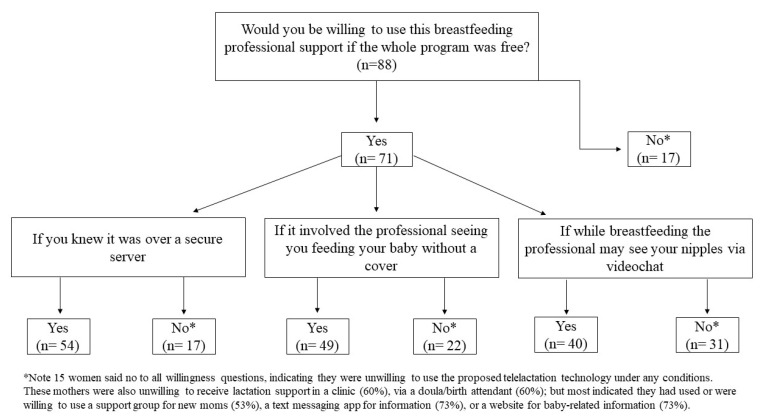
Conditions under which mothers would be willing to use telelactation services if it were free.

**Table 1 children-08-00271-t001:** Willingness to use telelactation survey questions.

Would You Be Willing to Use This Professional Breastfeeding Support Program If:
It cost $300 a month and there would be little or no wait times for help
It cost $210 a month and there might be a 5–10 min wait time for help
The first session was free
The professional spoke English but was from a different culture
The professional spoke English but lived outside the United States
Your doctor suggested it
It provided info on local resources
While breastfeeding the professional may see your nipples via videochat
It involved the professional seeing you feeding your baby without a cover
You knew it was over a secure server
The whole program was free

**Table 2 children-08-00271-t002:** Characteristics of the sample (*N* = 88).

Characteristics	
**Age (years, mean ± SD)**	29.3 ± 5.05
	*N* (%)
**Married**	48 (54.5)
**Working (full or part time or self-employed)**	29 (33.0)
**Income USD 50 K or more**	33 (37.5)
**Race/Ethnicity ***	
Black ^1^	22 (25.0)
White ^1^	45 (51.1)
Hispanic ^1^	22 (25.0)
**Has access to…**	
Desktop, laptop, netbook, or notebook computer	72 (81.8)
Tablet computer (iPad, Samsung Galaxy, Windows Tablet)	58 (65.9)
Mobile phone with text messaging	82 (93.2)
Smart mobile phone (iPhone or Android)	83 (94.3)
**Use the Internet to videochat (Skype, Oovo, FaceTime)**	47 (53.4)
**Willing (probably yes or definitely yes vs. other responses) to use telelactation if…**	
The whole program was free	71 (80.7)
You knew it was over a secure server	56 (63.6)
It involved the professional seeing you feeding your baby without a cover	49 (55.7)
While breastfeeding the professional may see your nipples via Videochat	40 (45.5)
**Beliefs (agree or strongly agree vs. other responses) about using mobile phone for telelactation**	
Videochat can provide access to breastfeeding support wherever women need it	57 (64.8)
If you use video-based breastfeeding support over a computer or Internetyou can never be sure who will have access	43 (48.9)
I do not feel comfortable with accessing video-based breastfeeding support online	23 (26.1)
Videochat is good for some things, but not for getting help with breastfeeding	19 (21.6)
Women will not develop a personal relationship with breastfeeding support professional	18 (20.5)
**Use of (or willingness to use) other lactation support services**	
Website for baby-related information	80 (90.9)
Text4Baby (a service that gives useful information to mothers via text)	67 (76.1)
Breastfeeding consulting at clinic	58 (65.9)
Breastfeeding help from a doula or traditional birth attendant	58 (65.9)
Support group for new moms	56 (63.6)

^1^ versus any other response. * Participants could select more than one option.

**Table 3 children-08-00271-t003:** Bivariate Associations between Participant Characteristics and Willingness or Beliefs Variables ^#^.

	Willing (Probably Yes or Definitely Yes vs. Other Responses) to Use Telelactation If…				Belief (Agree or Strongly Agree vs. Other Responses) about Using Mobile Phone for Telelactation	
	The Whole Program Was Free	You Knew It Was over a Secure Server	It Involved the Professional Seeing You Feeding Your Baby without a Cover	While Breastfeeding the Professional May See Your Nipples	Videochat Can Provide Access to Breastfeeding Support Wherever Women Need It	Women Will Not Develop a Personal Relationship with the Breastfeeding Support Professional
	OR (95% CI)	OR (95% CI)	OR (95% CI)	OR (95% CI)	OR (95% CI)	OR (95% CI)
Age	NS	NS	NS	NS	1.17 (1.05–1.30) **	NS
Married ^1^	3.69 (1.17–1.60) *	7.50 (2.79–20.15) ***	6.23 (2.46–15.79) ***	5.00 (1.99–12.60) **	4.20 (1.65–10.68) **	NS
Income ^2^	NS	4.03 (1.44–11.31) **	3.20 (1.26–8.13) *	3.32 (1.35–8.16) **	5.02 (1.69–14.92) **	NS
Black ^3^	NS	0.36 (0.14–0.98) *	0.35 (0.13–0.95) *	NS	NS	NS
White ^3^	NS	4.60 (1.79–11.83) **	3.77 (1.55–9.15) **	4.36 (1.73–10.45) **	2.69 (1.09–6.65) *	0.29 (0.09–0.90) *
Has used Videochat	NS	3.44 (1.28–8.55) **	3.68 (1.52–8.93) **	2.39 (1.01–5.67) *	NS	NS

NS = Non-significant, * *p* < 0.05, ** *p* < 0.01, *** *p* < 0.001. ^1^ versus not married, ^2^ income at least USD 50,000 per year versus less than USD 50,000 per year, ^3^ versus any other response. ^#^ Sociodemographic categories (working and Hispanic) as well as answers to belief questions (“you can never be sure has access to breastfeeding support services offered over the Internet;” “I do not feel comfortable accessing breastfeeding support online;” videochat is good for some things, but not for accessing breastfeeding support”) were non-significant in all bivariate analyses, and thus omitted from the table.

**Table 4 children-08-00271-t004:** Multivariable model: factors related to mothers agreeing or strongly agreeing to the statement “I do not feel comfortable with accessing video-based breastfeeding support online”.

Attributes	Full Model ^3^ OR (95% CI)	Final Model ^4^ OR (95% CI)
Willing to use telelactation if while breastfeeding the professional may see your nipples ^1^	0.19 (0.03–1.04)	0.20 (0.05–0.89) *
Willing to use telelactation if it involved the professional seeing you feeding your baby without a cover ^1^	1.18 (0.28–4.87)	--
Agree if you use video-based breastfeeding support over a computer or Internet you can never be sure who will have access ^2^	6.49 (1.83–23.06) **	6.32 (1.82–21.98) **
Agree videochat can provide access to breastfeeding support wherever women need it ^2^	0.28 (0.08–0.98) *	0.29 (0.08–0.99) *

* *p* < 0.05, ** *p* < 0.01. ^1^ probably yes or definitely yes versus any other response, ^2^ agree or strongly agree versus any other response. ^3^ Full model includes all the variables that were associated with the outcome in bivariate analysis. ^4^ Final model includes the variables that remained significant. The only variable eliminated through backward elimination was willingness to use telelactation if it meant feeding your baby without a cover.

**Table 5 children-08-00271-t005:** Key points of consideration regarding telelactation.

Provide anticipatory guidance upfront so mothers know what to expect during consultation
Assure mothers the sessions are not recorded
Adjust practice in consideration of mother’s comfort level
Think creatively about working around discomfort and technological capabilities (access to webcam or just cell phone; high speed internet via ethernet or using cellular network; creative use of mirrors to show additional angles)
Consider making videos using dolls or sending mothers other video resources with solutions to common breastfeeding problems (latch issues, milk production) to assist with education
Use HIPAA compliant and secure video conferencing platform, if available; providers should use private location with minimal background for appointment
Conduct sessions alone in private rooms and consider locking the door, to ensure no other person in the provider’s environment will compromise the mother’s privacy

## Data Availability

The data presented in this study are available on request from the corresponding author.

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
