# Peer review of "Modesty and Security: Attributes Associated with Comfort and Willingness to Engage in Telelactation"

_children, 2021, doi:10.3390/children8040271_

Round 1

Reviewer 1 Report

I was very pleased to read this very innovative and interesting paper on a so new, but very important topic. Congratulations on your work.

I just made some comments and suggestions directly in the pdf document, because I find it easier to read. 

Reviewer 2 Report

This is a study surveying mothers about their willingness to use a lactation consultant or receive lactation support services through telehealth. The biggest concern I had was that the ordering of questions asking about acceptability of using video chat may “prime” respondents.  In particular, describing women as exposing nipples as opposed to “feeding without a cover” may influence how women respond to these questions. Did the study team vary the order of these questions? If yes, they could examine whether this is a concern by looking at answers for women who are asked about nipples first as compared to women who are asked about this last.

Other comments:

114 women consented and provided survey – can you say how many were approached? How do those who completed the survey compare to those who did not?  And the 88 analyzed – why did you exclude 26 and how do they compare to the 114 and the broader population of women delivering at these hospitals?

Figure 1 makes it look like you only asked about comfort among those who were willing to use if it was free? Is that true? If not, why present like this?

Are you powered to detect relationships among the 26% of participants who were uncomfortable (Table 3)?
